# Molecular Diagnostic Tools against SARS-CoV-2 in Poland in 2022

**DOI:** 10.3390/biomedicines10123259

**Published:** 2022-12-15

**Authors:** Lukasz Fulawka, Aleksandra Kuzan

**Affiliations:** 1Molecular Pathology Centre Cellgen, 50-353 Wroclaw, Poland; 2Department of Biochemistry and Immunochemistry, Wroclaw Medical University, 50-367 Wroclaw, Poland

**Keywords:** COVID-19, SARS-CoV-2, diagnostics, false-negative results, PCR

## Abstract

The most effective way to stop the spread of COVID-19 (coronavirus disease 2019) is to detect severe acute respiratory syndrome coronavirus 2 (SARS-CoV-2) and isolate those infected as soon as possible. More than 1000 types of molecular and antigen-based immunoassay tests to detect SARS-CoV-2 are now commercially available worldwide. In this review, we present the possibilities of molecular diagnostics available in Poland in 2022. We provide a description of what samples have proven useful to confirm SARS-CoV-2 infection, we describe what methods are used, as well as what safeguards can and should be used to prevent false-negative and false-positive results, and finally we review the products that diagnostic laboratories have to choose from. We also describe diagnostic problems associated with the mutation of the virus.

## 1. Introduction

SARS-CoV-2 is considered a new human coronavirus that was identified in 2019, and it is genetically different from common human coronaviruses (229E, NL63, OC43, and HKU1) that cause seasonal acute respiratory diseases. It is also genetically different from the two newer human coronaviruses, MERS-CoV (Middle East respiratory syndrome coronavirus) and SARS-CoV [1].

The SARS-CoV-2 epidemic has undoubtedly caused extensive damage to societies and the economy. However, the epidemic seems to have allowed molecular biology to develop quite rapidly. Many companies intensively developed a variety of tests, which quickly obtained IVD (in vitro diagnostic) status. In this review, we aimed to analyze what tests are available on the Polish market 3 years after the outbreak of the pandemic.

The symptoms of COVID-19 are quite nonspecific, similar to influenza or other infectious diseases; hence, there is a need to perform diagnostic tests to confirm SARS-CoV-2 infection. In this paper, we focus on molecular tests as they have been endorsed as the gold standard by the WHO (World Health Organization) [1]. The sample types that were considered for the diagnosis of SARS-CoV-2 are listed below.
Nasopharyngeal swabs, with a relatively very high positive rate of 92.2%; the disadvantage is that sampling is uncomfortable for patients [2];Oropharyngeal swabs, with a lower positive rate than in sputum, but higher than nasopharyngeal swabs [3]; it has a sensitivity of 84% [2];Sputum, with the highest sensitivity-97.2% [4] and the highest positive rate for COVID-19 patients with different degrees of disease severity; however, only a small portion (28–33.7%) of COVID-19 cases show sputum production [3];BALF (bronchoalveolar lavage fluid), with an improvement of diagnostic accuracy in severe cases; however, it requires both a suction device and a skilled operator, and it is also a painful process for the patients. Examination of BAL fluid has been shown to have a sensitivity of 90–100% [3];Rectal swabs or stools, with positive rates varying from 29% to 83.3% [3]; such a wide range of the value has resulted in this method not being adopted in Poland;Saliva, with ease of sample collection, no interaction during sampling, and high viral titers during early stages of infection [5]; it has a sensitivity close to 84% and is potentially more sensitive than nasopharyngeal samples when using crude samples [6];Blood, with a highly unsatisfactory sensitivity of 7.3% [4]. This result implies that COVID-19 diagnostics should not be performed on blood samples; however, since it is stated that viremia in COVID-19 may occur, nurses and diagnosticians should be particularly careful when working with the blood of infected patients);Urine is unsuitable for diagnostics, with a sensitivity of tests on this material close to 0% [4].

## 2. Method Types

WHO presents NAATs (nucleic acid amplification tests) as a reference [1]. The classic tests rely on detecting the presence of viral RNA by reverse transcription polymerase chain reaction (RT-PCR). Multiple strategies exist for amplifying target genes, including reverse transcriptase loop-mediated isothermal amplification (RT-LAMP), recombinase polymerase amplification (RPA), helicase-dependent amplification (HDA), strand displacement amplification (SDA), and nucleic acid sequence-based amplification (NASBA) [7]. Recently, assays based on the isothermal amplification of viral nucleic acids have been developed, in combination with clustered regularly interspaced short palindromic repeat (CRISPR)-based detection methods [1]. Below is a brief description of the most popular of the abovementioned methods.

### 2.1. RT-qPCR

Real-time/quantitative reverse transcription polymerase chain reaction (RT-qPCR) is a relatively simple method that allows detecting the presence of a specific fragment of genetic material. After isolating the RNA from the sample, a reverse transcriptase reaction is performed, and then the sequences of the appropriate genes are amplified using their specific primers. Detection of amplicons is possible thanks to the use of fluorescent dyes, with the growth of products being monitored in real time. Real-time RT-PCR remains the most popular molecular method in the diagnosis of COVID-19 in Poland and Europe [8,9]. Its advantages are its simplicity, high accessibility, high sensitivity, and high specificity, as well as the fact that the tests based on this method can be quantitative. 

### 2.2. Isothermal Amplification Methods

These methods, which do not require thermal cycling, are more rapid than RT-PCR, characterized by comparable sensitivity and specificity, and they are also considered suitable as point-of-care tests for the detection of SARS-CoV-2 [1]. 

#### 2.2.1. RT-LAMP (Reverse Transcription Loop-Mediated Isothermal Amplification)

RT-LAMP uses a combination of 4–6 primers against different loci of target DNA sequences with the constant reaction temperature varying between 60 and 65 °C. Reverse transcription is performed, followed by loop-mediated isothermal amplification. The advantage is the shorter time, which can only take 30 min, as well as the lack of the need for specialized equipment such as thermal cycler [5,10].

#### 2.2.2. CRISPR (Clustered Regularly Interspaced Short Palindromic Repeats)

The principle of this method is the activity of endoribonuclease Cas with an sgRNA (single guide RNA) complex followed by probing of the amplified specific fragments of the virus. The test can be designed according to the measurement of fluorescence or lateral flow [10,11]. COVID-19 diagnosis uses the Cas-13 enzyme, which has the ability to react with RNA; after reverse transcription, the Cas12 or Cas9 enzyme possibly interacts with cDNA [10]. The advantage of Cas12 and Cas13 over Cas9 is that they have collateral activity, making these enzymes more likely to be used in diagnostics [12].

### 2.3. Genomic Sequencing

Genomic sequencing of SARS-CoV-2, mostly based on next-generation sequencing (NGS), is used to identify pathogens through sequencing of nucleic acid fragments, followed by analyzing and comparing biological information and databases with high accuracy [7]. The method is generally used for genetic mapping rather than diagnostic testing due to its high cost and long detection time [7].

### 2.4. An Alternative to NAATs

An alternative to NAATs is to use tests detecting antigens or antibodies.

The sensitivity of an antigen test is strongly dependent on the viral load. If it is high (Ct < 25 on a real-time RT-PCR test), the sensitivity can reach up to 100%. If, on the other hand, the viral load is low (Ct > 35 on a real-time RT-PCR test), the sensitivity of the antigen test can be as low as 22.2%, i.e., more than two-thirds of results may be false negatives [13]. 

Immunoassays detecting antibodies against SARS-CoV-2 have the major disadvantage that they cannot be used to detect an infection, but only confirm that there has been a past immune system response to the pathogen. The sensitivity of the immunoassay strongly depends on the time elapsed since the infection, on the patient’s individual characteristics, and on the biochemical features of individual diagnostic kits. One study comparing five tests for IgG antibodies showed that the sensitivity of serum samples collected on average 24 days after the onset of symptoms was between 74.5% and 88.2% [14]. Earlier, IgM antibodies appear. It should be remembered that both disease and vaccination cause different immune system reactions in each patient; therefore, a negative result of such a test does not necessarily mean that the patient has not been infected.

The great advantage of antigen and serological tests is that the test can be relatively easily designed to be a so-called rapid diagnostic test (RDT) or a point-of-care test (POCT), to be performed quickly, without specialist equipment or training [15]. 

### 2.5. Diagnostic Windows

It should also be borne in mind that the diagnostic window for molecular, antigen-based, and antibody-based methods is quite narrow. The narrowest is for antigen tests, whereas it is relatively wide for molecular tests, and the broadest window exists for tests detecting IgG [15]. In general, RT-PCR tests show the highest sensitivity between days 5 and 14 after contact with the SARS-CoV-2 coronavirus, and this is usually the time when a patient experiences the first symptoms of infection. Thus, it is sometimes possible to achieve a false-negative molecular test result for SARS-CoV-2 due to the test being performed before the virus reaches a detectable number of copies in the epithelial cells. Likewise, if the test is carried out too late, the opposite may happen, as false-positive results may be obtained because the RNA of the virus can remain in the epithelium for a long time (see Section 6). Individual differences are important here, because the immune system will react differently in each patient; hence, the dynamics of virus replication will be different [15]. In some patients infected on the same day, the molecular test may be positive after 2 days, whereas it may take longer in others.

## 3. Diagnostic Errors and Ways to Prevent Them

It is worth emphasizing the fact that even the most sensitive and specific molecular methods are not without the possibility of making a mistake and obtaining a false result. The false-negative rate of RT-PCR varies from 3% to 41% [16]. It is estimated that around 5% of test results may be false positives [17]. The types of errors are presented in Table 1, along with the steps that should be taken to minimize the probability of obtaining false results.

The strategy of avoiding false positives involves performing additional positive and negative controls. The positive control, i.e., carrying out a PCR reaction with a sample of the isolated virus RNA, necessarily on each plate/strip, parallel to the samples taken from patients, allows verifying that the amplification and detection are correct. A negative control, generally called a “no template control” (NTC), is a reaction where buffer or water is added instead of the sample to eliminate contamination of the reagents and the laboratory work zone in general. Internal controls (ICs) can be divided into exogenous and endogenous controls. The first represents a housekeeping gene, i.e., a gene that is constitutively expressed in all human cells. Typically, it is gene of β-actin (*ACTB*), glyceraldehyde-3-phosphate dehydrogenase (*GADPH*), RNAse P, or β-globin. Detection of ICs verifies that the sample contains human mRNA and that PCR takes place properly. Internal exogenous controls are generally synthetic nucleic acids added to each sample before the extraction process, which check extraction, reverse transcription, and PCR [19]. Providing an internal control is not obligatory, but generally guaranteed by manufacturers of diagnostic kits. This significantly increases the reliability of the results. Appendix A) presents information on what types of controls are included in the tests. Figure 1 graphically illustrates what proportion of all tests includes an internal control (only slightly more than a half). Of the 65 kits analyzed, only three had both an exogenous and an endogenous control. The most common gene analyzed as part of the endogenous control was the gene for RNAse P, with as many as 13 out of 22 sets containing primers for this sequence. The remaining kits analyzed beta-actin mRNA (3/22), analyzed other sequences, or did not specify what sequence. It has to be underlined that almost all kits that did not provide ICs at all were reflex tests, i.e., secondary tests conducted after initial positive results.

## 4. What Genes Are Analyzed in PCR for the Detection of SARS-CoV-2?

The SARS-CoV-2 genome is nearly 30,000 nucleotides long, making it one of the largest RNA viruses. The longest fragment here is organized into two overlapping ORFs (open reading frames), ORF1a (11–13 kb) and ORF1b (7–8 kb), which codes for RNA-dependent RNA polymerase (RdRp). Its sequence is highly specific; therefore, it is often chosen for detection in PCR tests for COVID-19 diagnostics [1]. The first RT-PCR protocols for the detection of SARS-CoV-2 targeted genes related to RdRp, nucleocapsid (N), and envelope protein (E) [18]. The tests based on the detection of RdRp used two probes; one probe called “Pan Sarbeco-probe” detected bat-related SARS coronaviruses and other coronaviruses, while the other probe called RdRp-p2 was specific only for SARS-CoV-2 [18]. Currently, tests are being developed to detect very diverse sequences of the entire genome of the SARS-CoV-2 virus, in addition to the abovementioned sequences for RdRp, the spike protein (S), and the membrane (M) protein [20].

To minimize the risk of false positives related to the presence of non-SARS-CoV-2 virus, WHO recommends the detection of at least two different targets on the COVID-19 virus genome. Despite the large proportion (38% of commercially available tests) being based on analyzing only one gene, most of them are reflex tests, detecting specific variants in positive samples. Only a few of them were developed in the beginning of the pandemic and are still available, but they are generally not used in Europe. On the other hand, to increase the reliability of the test, some companies have developed tests that analyze up to three or four genes (see Figure 2). Appendix A shows which genes are detected by the analyzed kits. It turns out that the most frequently chosen target is the ORF1ab gene, with several tests also analyzing the S protein gene and the nucleocapsid gene (Figure 3).

## 5. Do Diagnostic Kits Detect Current SARS-CoV-2 Variants?

SARS-CoV-2, like other RNA viruses, is prone to genetic evolution. Even a single-nucleotide substitution can cause a change in virulence properties relative to the parent virus variance [21]. The changes may concern the interaction of the virus with host proteins or the ability to avoid mechanisms of the host’s immunity, which undermines the effectiveness of vaccination. In addition, changes can also cause problems in the diagnosis of infections, because molecular tests may not detect altered sequences [22].

During the pandemic, many variants have been distinguished, i.e., variants of concern (VOCs): (1) Alpha (B.1.1.7; September 2020); (2) Beta (B.351, B.1.351.2, and B.1.351.3 in December 2020); (3) Delta (B.1.617.2, AY.1, AY.2, AY.3, and AY.3.1 in December 2020); (4) Gamma (P.1, P.1.1, and P.1.2 in January 2021); Omicron (B.1.1.529 in November 2021) [22]. In addition, several variants of interest (VOIs) have arisen, e.g., Zeta (P.2), Kappa (B.1.617.1), and Mu (B.1.621 and B.1.621.1). In those cases, specific genetic markers were found that are predicted to affect transmission, diagnostics, therapeutics, or interaction with the immune system [22]. 

The Omicron variant is an example of a variant with multiple mutations, from which the potential diagnostic problems that may arise in the future can be extrapolated. The mentioned variant has over 30 mutations in a key gene for the S protein, resulting in a 13-fold increase in viral infectivity and substantial escape from neutralizing antibodies induced by vaccination [22,23]. It can also be suspected that some tests based on the detection of the S protein gene will show false-negative results for the Omicron variant. Analyses in the direction of changes in the test effectiveness depending on the virus mutation can be carried out experimentally, as well as in silico.

Metzger et al. analyzed 39 test kits commercially available in Switzerland and Liechtenstein, and they exchanged kits for S protein sequences so as to address the potential of not detecting Omicron. Only two of the eight assays targeting the S gene appeared to show S-gene dropout with the Omicron variant. However, it is emphasized that these data are preliminary, based on in silico experience, and that each set should be thoroughly evaluated [24]. Without the possibility of verifying the effectiveness of all tests aimed at the S protein available in Poland, we would like to point out that as many as 35 tests out of 65 available in Poland are based on the detection of the S protein gene (see Appendix A). Fortunately, many of them also detect other genes. It is worth reminding that, in order to minimize the risk of not detecting a mutant version of the virus (Omicron or another), it becomes necessary to search for at least two molecular targets in one PCR test, i.e., independent genes of a given virus.

## 6. Recurrence of Positive SARS-CoV-2 in Patients Recovered from COVID-19

There are relatively many cases when a patient, shortly after recovering from COVID-19, is again positive for the presence of the SARS-CoV-2 virus according to a molecular test. This is very worrying as it may indicate a lack of immunity and a further infection with the same virus. Possible causes are also false RT-PCR results, intermittent virus shedding, viral reactivation, reinfection with another SARS-CoV-2 strain, or exposure to a contaminated environmental surface after recovery [16]. The paper by Dao et al. (2021) listed arguments for each of these possibilities. As PCR does not distinguish between live and dead viruses, cultivation tests should be performed to confirm that the virus is inactivated. If it turns out that the virus is alive and active after the time of remission, sequencing should be performed to see if it is a reinfection or activation of the surviving virus in the patient’s body. It must be mentioned that cultivation of viruses is a method used in sophisticated research rather than diagnostics. It seems that patients who obtain positive PCR tests shortly after infection are not infectious [16]. According to preliminary data, the Omicron variant of SARS-CoV-2 has a higher risk of reinfection [22].

## 7. Additional Advantages of Some Tests: Multiplexing—Detection of Other Pathogens in One Test

Influenza, COVID-19, and other infectious agents that cause so-called “colds” have many common features; they attack the respiratory tract, causing, among other symptoms, fever, cough, general weakness and breakdown, pain in muscles and joints, sore throat, and, less often, diarrhea. Due to the similarity of the symptoms of these diseases, with significant differences in further medical management, genetic testing of what virus causes the symptoms is very valuable. Many companies have developed multiplex tests to detect what type of virus has infected a patient. The analysis of Appendix A allows identifying such products that, in addition to SARS-CoV-2, allow the detection and differentiation of influenza A and B virus or respiratory syncytial virus (RSV) type A or B. “Record-holders” in this area have prepared tests with a panel of 22 pathogens; in addition to SARS-CoV-2, the panel allows the detection of influenza A virus, influenza A virus H1N1/2009, influenza A virus H1, influenza A virus H3, influenza B virus, coronaviruses HCoV-229E, HCoV-HKU1, HCoV-NL63, and HCoV-OC43, parainfluenza virus type 1, parainfluenza virus type 2, parainfluenza virus type 3, parainfluenza virus type 4, RSV type A/B, human metapneumovirus type A/B, adenovirus, bovavirus, rhinovirus/enterovirus, Mycoplasma pneumoniae, Legionella pneumophila, and Bordetella pertussis. To perform such a test, a small sample volume is required, a minimum amount of time is spent, and the results there are available in approximately 1 h. However, this requires appropriate equipment dedicated to the reagent cassettes and is generally costly [25].

## 8. Comparison of Diagnostic Products Dedicated to the Detection of SARS-CoV-2 Available in Poland to Those Available in Other Countries

Similar combinations of molecular tests for SARS-CoV-2 are performed in many countries in order to be able to compare the possibilities of diagnostic companies and what new products appear on local diagnostic markets. For the purposes of this manuscript, we aimed to compare the description of possibilities in Poland to those described in China [7], India [18], and Italy [1]. The pandemic started in China, and the biotechnology industry there is very buoyant; thus, it is not surprising that the number of diagnostic products for SARS-CoV-2 from China, especially from the Wuhan area, is very large. The paper by Ruhan et al. (2020) listed 24 tests produced domestically, not only of the RT-PCR type, but also, e.g., tests based on isothermal amplification and gold probe-based chromatography [7]. Therefore, the Chinese market seems to be independent. European markets often import Chinese products. In Italy, for example, six out of 41 presented tests were of Chinese origin; generally, a large share of tests were of Asian origin, and none were of Italian origin [1]. In Poland, we also have two tests from China available, as well as tests from various European and non-European countries, including those produced in Poland. It is interesting to note that some products can be found in Italian, Indian, and Polish diagnostic publications. In our list, we intentionally only mention tests based on RT-PCR, as Appendix A is very extensive. Nevertheless, it can be seen that the diversity is increasing compared to studies from 2020 and 2021.

## 9. Summary and Conclusions

More than 1000 types of molecular and antigen-based immunoassay tests to detect SARS-CoV-2 are now commercially available worldwide [15]. This review of RT-PCR tests for COVID-19 diagnostics was primarily aimed at showing the diversity that has been achieved in a relatively short time, which was unprecedented in history. Through an analysis of Appendix A, the reader can decide the most appropriate test. Attention to the need for appropriate control and detection of more than one gene will certainly improve the effectiveness of diagnostics.

There are many indications that the pandemic is ending, but qPCR is and will be needed in a wider context. qPCR is the method of choice for the diagnosis of other viral diseases, such as viral hepatitis (HAV, HBV, and HBV), AIDS (HIV), human papillomavirus (HPV), cytomegalovirus (CMV), or infections caused by atypical bacteria, such as Chlamydia or Mycoplasma. The lesson that biotechnology companies have received from COVID-19 is invaluable, and we believe that it will improve diagnostics in Poland and around the world.

## Figures and Tables

**Figure 1 biomedicines-10-03259-f001:**
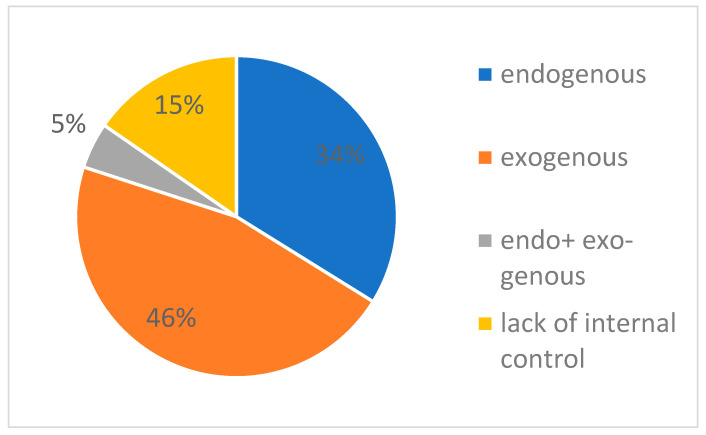
The type of internal control used in PCR tests for SARS-CoV-2 available in Poland.

**Figure 2 biomedicines-10-03259-f002:**
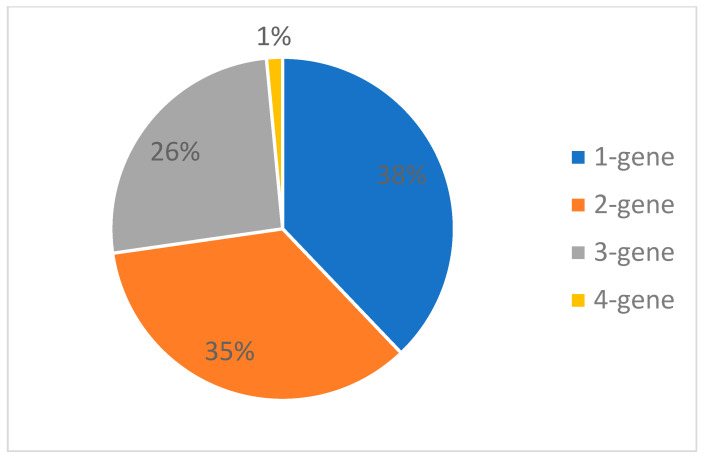
The number of genes detected by PCR tests for SARS-CoV-2 available in Poland.

**Figure 3 biomedicines-10-03259-f003:**
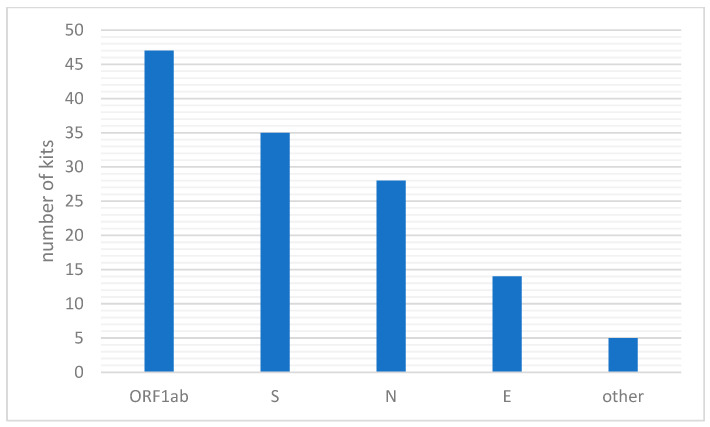
Illustration comparing SARS-CoV-2 genes in terms of the frequency of their detection in PCR-type tests available in Poland.

**Table 1 biomedicines-10-03259-t001:** Comparison of diagnostic errors, along with their effects and countermeasures [7,11,16,17,18,19].

	Preanalytical Phase	Analytical Phase	Post-Analytical Phase	Prevention
False-negative results	-Incorrect sampling -Too long storage -Inappropriate storage conditions	-Incorrect extraction technique -Pipetting errors -Abnormal RT-PCR reaction conditions (temperature, reagent concentrations) -Using incorrectly stored reagents -Target mutation	-Misinterpretation -Incorrect determination of baseline and threshold	-Positive controls (viral RNA sequence) -Internal controls, e.g., endogenous (housekeeping genes) and exogenous (artificial construct)
False-positive results	-Sample contamination at the collection point	-RNA contamination during extraction -Use of contaminated reagents -Cross-reaction with other viruses (e.g., other coronaviruses)	-Misinterpretation -Incorrect determination of baseline and threshold	-Negative controls (with no template) -Test based on 2 or 3 viral genes

## Data Availability

Not applicable.

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
