# Peer review of "Molecular Diagnostic Tools against SARS-CoV-2 in Poland in 2022"

_biomedicines, 2022, doi:10.3390/biomedicines10123259_

Round 1
Reviewer 1 Report
In this review article, the authors intend to describe the molecular diagnostics available in Poland in 2022.
Several suggestions:
1. It is better to have references after the paragraphs in lines 24, 140, 172, 197, 209,224, 272, 292.
2. Please mention the references for Table1, Figures 1, 2, and 3.
3. (severe acute respiratory syndrome coronavirus 2) in line 10 should be removed to line 9 at its first appearance.
4. It should be [real-time RT-PCR] not [RT-PCR] in lines 79, 109, 110.
5. Please check [proteinases Cas], [proteinases]?
Author Response
We would like to thank the Reviewer for the time devoted to the analysis of this work, all valuable comments and the opportunity to improve the work. Below the answers "point by point".
- It is better to have references after the paragraphs in lines 24, 140, 172, 197, 209,224, 272, 292.
We apologize for these oversights. We supplemented the citations in all the indicated places, except for the verse 209- it was our own conclusion based on observations, so we are unable to provide the literature source.
- Please mention the references for Table1, Figures 1, 2, and 3.
For table 1 we have added literature sources. For Figures 1-3 it is impossible - the data come from our own study of the Polish biotechnological and diagnostic market. The full data used to draw up Figures 1-3 are presented as the supplement file in the form of a table (S1).
- (severe acute respiratory syndrome coronavirus 2) in line 10 should be removed to line 9 at its first appearance.
We apologize for the error, we have corrected it.
- It should be [real-time RT-PCR] not [RT-PCR] in lines 79, 109, 110.
We apologize for the error, we have corrected it.
- Please check [proteinases Cas], [proteinases]?
We apologize for the error, we have corrected it (proteinases à proteases).
Thanks again to the Reviewer for all suggestions, we hope that we have responded satisfactorily and that the manuscript will be accepted for publication in Biomedicines in Special Issue „State-of-the-Art Molecular and Translational Medicine in Poland”.
Reviewer 2 Report
The authors present an interesting and timely review article on Molecular diagnostic tools for SARS-CoV-2 in Poland. I consider the manuscript appropriate for the journal, specially for the special issue it is submitted to. However, some minor comments should be resolved before publication:
- General comments:
- References should be added throughout the manuscript. There is a clear shortage of references in most parts of the review.
- Minor spelling mistakes and erratum should be corrected. Please carefully review the manuscript.
- Line 36: nasopharyngeal samples can be obtained without the supervision of a professional.
- Line 52: could be interesting to mention that saliva can be more sensitive than nasopharyngeal samples, when using crude samples.
- Line 81: Can be RT-qPCR really considered simple when it needs complex equipment and professionals to performe it?
- Line 83: Maybe the section title could be: "Isothermal amplification methods", more appropiate than just "Isothermal".
- Line 91: Please, review english.
-Line 97: Erratum, please correct to Cas-9. Additionally, Cas-13 should also be mentioned in this section and the advantages of collateral activity of both Cas-12 and Cas-13 over Cas-9 that does not have it for diagnostic purposes.
-Table 1: Please, modify the format so the Table can be read more easily.
- Line 176: Erratum, please correct to 30,000 bases
- Line 213: If all VOC are numbered, Omicron should be numbered too.
- Line 238: Please review english language on this phrase.
Author Response
We would like to thank the Reviewer for the time devoted to the analysis of this work, all valuable comments and the opportunity to improve the work. Below the answers "point by point".
- References should be added throughout the manuscript. There is a clear shortage of references in most parts of the review.
We apologize for this error. Literature references have been added in many places, we hope that the current informations in the manuscript are cited correctly and legibly.
- Minor spelling mistakes and erratum should be corrected. Please carefully review the manuscript.
The manuscript was checked, in questionable places asking English-speaking colleagues for their opinion on correctness. We hope that we have removed all errors.
- Line 36: nasopharyngeal samples can be obtained without the supervision of a professional.
In health care facilities during the pandemic in Poland, it was recommended that nurses or doctors take swabs, because patients usually inserted swabs too shallowly. However, the statement was deleted because it is indeed possible for the patient to take the swab himself.
- Line 52: could be interesting to mention that saliva can be more sensitive than nasopharyngeal samples, when using crude samples.
Thank you for this suggestion, we have added this information.
- Line 81: Can be RT-qPCR really considered simple when it needs complex equipment and professionals to performe it?
Compared to sequencing or other molecular methods, it is a fairly simple method, first of all a method well known to diagnosticians, unlike LAMP and others; however, the notation was softened by adding "relatively" simple.
- Line 83: Maybe the section title could be: "Isothermal amplification methods", more appropiate than just "Isothermal".
Thank you for this advice, we have completed this subtitle to make the name full.
- Line 91: Please, review english.
Verified with an English speaking teacher.
-Line 97: Erratum, please correct to Cas-9. Additionally, Cas-13 should also be mentioned in this section and the advantages of collateral activity of both Cas-12 and Cas-13 over Cas-9 that does not have it for diagnostic purposes.
We apologize for the error in this part and thank you for suggesting how to complete this paragraph. Corrected according to comments.
-Table 1: Please, modify the format so the Table can be read more easily.
The table has been extended to full width, we hope it is clear enough now.
- Line 176: Erratum, please correct to 30,000 bases
We apologize for this error, thank you for pointing it out, we have corrected it.
- Line 213: If all VOC are numbered, Omicron should be numbered too.
Unfortunately, we are not sure what Reviewnet means, after all, the number is given: Omicron (B.1.1.529).
- Line 238: Please review english language on this phrase.
Verified with an English speaking teacher.
Thanks again to the Reviewer for all suggestions, we hope that we have responded satisfactorily and that the manuscript will be accepted for publication in Biomedicines in Special Issue „State-of-the-Art Molecular and Translational Medicine in Poland”.
Round 2
Reviewer 1 Report
1. Please check [proteases Cas] again in line 98, Is Cas an endoribonuclease?
Author Response
- Please check [proteasesCas] again in line 98, Is Cas an endoribonuclease?
--> We thank the reviewer for consistently pointing out this error. Cas proteins are indeed endonucleases, not proteases, although some publications use the phrase „proteases Cas”, hence our error. We apologize for it and thank you for verifying the correctness of our manuscript and for the care with which the Reviewer performed it.